# "We're very much part of the team here": A culture of respect for Indigenous health workforce transforms Indigenous health care

Emma V. Taylor[1]*, Marilyn Lyford[1], Lorraine Parsons[1], Toni Mason[2], Sabe Sabesan[3], Sandra C. Thompson[1]

1 Western Australian Centre for Rural Health, The University of Western Australia, Geraldton, Western Australia, Australia, 2 Aboriginal Health Unit, Mission, St Vincent's Hospital Melbourne, Fitzroy, Victoria, Australia, 3 Townsville Cancer Centre, Townsville Hospital and Health Service, Douglas, Queensland, Australia

* emma.taylor@uwa.edu.au

## Abstract

### Background

Improving health outcomes for Indigenous people by strengthening the cultural safety of care is a vital challenge for the health sector, both in Australia and internationally. Although Indigenous people have long requested to have Indigenous practitioners involved in their health care, many health services report difficulties with recruiting and retaining Indigenous staff. This article describes Indigenous workforce policies and strategies from two Australian health services, as well as cancer-service specific strategies.

### Methods

Services were identified as part of a national study designed to identify and assess innovative services for Indigenous cancer patients and their families. In-depth interviews were conducted in a small number of identified services. The interviews from two services, which stood out as particularly high performing, were analysed through the lens of Indigenous health workforce.

### Results

Twenty-four hospital staff (Indigenous and non-Indigenous), five Indigenous people with cancer and three family members shared their views and experiences. Eight themes were identified from the way that the two services supported their Indigenous workforce: strong executive leadership, a proactive employment strategy, the Indigenous Health Unit, the Indigenous Liaison Officer, multidisciplinary team inclusion, professional development, work environment and a culture of respect. Participants reported two positive outcomes resulting from the active implementation of the eight workforce themes: 'Improved Indigenous patient outcomes' and 'Improved staff outcomes'.

**Data Availability Statement:** The qualitative dataset analysed during the current study cannot be shared publicly due to the risk of compromising

the confidentiality of participants and violating the agreement to which the participants consented. De-identified data is available from the WA Centre for Rural Health (contact via +61 08 9956 0200 or via admin-wacrh@uwa.edu.au) for researchers who meet the criteria for access to confidential data.

**Funding:** This study was undertaken as a component of the Centre of Research Excellence in Discovering Indigenous Strategies to improve Cancer Outcomes Via Engagement, Research Translation and Training (DISCOVER-TT CRE, funded by the National Health and Medical Research Council #1041111), and the Strategic Research Partnership to improve Cancer control for Indigenous Australians (STREP Ca-CIndA, funded through Cancer Council NSW (SRP 13-01) and the Cancer Council WA). The funders had no role in study design, data collection and analysis, decision to publish or preparation of manuscript.

**Competing interests:** The authors have declared that no competing interests exist.

## Conclusions

These two cancer services and their affiliated hospitals show how positive patient outcomes and a strong Indigenous health workforce can be achieved when a health service has strong leadership, commits to an inclusive and enabling culture, facilitates two-way learning and develops specific support structures appropriate for Indigenous staff. It is hoped that the strategies captured in this study will be used by health services and cancer services to inform their own policies and programs to support building their Indigenous workforce.

## Introduction

Aboriginal and Torres Strait Islander peoples are strong and resilient, with richly diverse and complex cultures which have developed over 50,000 years. Representing 3.3% of the total Australian population, Aboriginal and Torres Strait Islander peoples reside across all Australian states and territories [1]. However, colonisation had a devastating impact on traditional lifestyles, while systemic discrimination and disadvantage means that Australia's First Peoples continue to experience lower levels of education and employment, poorer health and shorter life expectancy [2–4]. While many First Australians enjoy good health, the disparity in outcomes between Aboriginal and Torres Strait Islander peoples and non-Indigenous Australians across a wide range of diseases, including cancer, has been widely reported for decades [4–6]. Although there have been reductions in the Indigenous mortality rate from chronic diseases, Indigenous mortality rates from cancer are rising and the gap in cancer mortality rates is widening [4, 7, 8]. (The term 'Indigenous Australians' is respectfully used hereafter to refer to Australia's Aboriginal and Torres Strait Islander peoples, while acknowledging the tremendous diversity of cultures and experiences of Australia's First Peoples.)

A number of factors contribute to the disproportionate levels of poor health experienced by Indigenous Australians, including socioeconomic disadvantage; lack of culturally appropriate health services and lower access to health services; and biomedical, behavioural, and environmental factors [2, 9, 10]. Indigenous Australians are more likely to have lifestyle and behavioural risk factors for cancer such as smoking, poor nutrition and physical inactivity [11]. However, reasons for poorer outcomes for Indigenous cancer patients are multifactorial and include lower participation rates in screening programs; later stage at diagnosis; the presence of comorbidities; lower uptake and completion of cancer treatment; and entrenched racism in the health system [8, 12–14]. Barriers to the use of health services by Indigenous people include fear or lack of trust of mainstream health facilities and lack of respect or understanding shown by health care providers [15, 16]. Research has demonstrated that the accessibility and cultural appropriateness of a health service has a significant impact on whether Indigenous people are willing to present for diagnosis and continue with treatment [17, 18]. Therefore, health service providers have an important role to play in encouraging Indigenous people to attend, thereby improving outcomes for Indigenous people, including those suffering from cancer. One way this can be achieved is through greater Indigenous representation in the health workforce generally, and more specifically in those providing cancer treatment and support.

Indigenous people have repeatedly expressed their desire to have Indigenous practitioners involved in their health care [16, 18–20]. Indigenous health professionals (defined as doctors, nurses, allied health professionals, health workers and liaison officers) understand the needs and priorities of Indigenous patients and can help them feel safer and more comfortable while

receiving care [21]. Research has demonstrated that Indigenous health professionals improve outcomes for Indigenous patients through advocacy, improving treatment attendance and compliance, and reducing discharges against medical advice; in addition they help non-Indigenous clinicians provide more culturally appropriate care [22–24]. However, Indigenous Australians are under-represented in the health workforce, and many health services report difficulties with recruiting and retaining Indigenous staff [25–27]. There are suggestions that the problem is growing, with organisations providing Indigenous primary health services (such as Aboriginal Community Controlled Health Services) reporting a 59% increase in the number of vacant FTE positions over 5 years [27, 28]. Furthermore, some Indigenous health professionals chose not to self-identify in mainstream organisations to avoid discrimination or being given responsibility for everything 'Indigenous' [29, 30]. Health services require policies and strategies to recruit and retain Indigenous staff, yet the literature reports few effective interventions around supporting an Indigenous health workforce [30–32].

This article presents Indigenous workforce strategies from two cancer services and their affiliated hospitals. The services were identified in a multi-stage, national study as particularly high performing and innovative in their provision of cancer services for Indigenous cancer patients and their families [33, 34]. The cancer services are both within public tertiary teaching hospitals, however are vastly different with respect to rurality, management and patient cohort, as summarised in Table 1.

Although our research focussed around Indigenous cancer care, patient care is delivered by staff and therefore workforce is critically important. Given the importance of cultural safety and the well documented wishes of Indigenous people to have Indigenous staff involved in their care [16, 18, 19], we explore here how these high performing services have adopted a "whole of service" approach to developing a stable and effective Indigenous health workforce. This article describes policies and strategies from both hospitals, as well as the cancer-service specific strategies.

## Methods

The research project was a component of research within the Centre for Research Excellence (CRE), Discovering Indigenous Strategies to improve Cancer Outcomes Via Engagement, Research Translation and Training Centre of Research Excellence (DISCOVER-TT). The CRE was led by an Indigenous researcher, and involved Indigenous and non-Indigenous people working together to improve services and outcomes for Indigenous people with cancer.

**Table 1. Characteristics of participating health services.**

|  | Urban Service | Regional Service |
|---|---|---|
| Location | Major capital city | Large regional centre |
| Management | Private | State health service |
| Size | 900 beds | 800 beds |
| Total staff | 5,700 | 6,000 |
| Indigenous staff (percent of total) | 52 (0.9%) | 241 (3.74%) |
| Catchment | Metropolitan-based, 11% rural | 670,000 people across 148,000 km$^2$ |
| Indigenous proportion of catchment population | 0.8%[a] | 8% |

[a] Proportion of population of the state.

This study forms part of a broader investigation to identify and describe cancer services providing treatment to Indigenous cancer patients in Australia. Public cancer treatment centres across Australia were surveyed to identify the type of cancer services provided, their Indigenous patient numbers and explore policy and implementation of Indigenous-specific initiatives [33]. Surveys were completed for 58 of 125 public cancer treatment centres. Based on findings from the survey, follow-up interviews were conducted with 14 service providers to explore current practice and programs aimed at improving cancer care for Indigenous Australians [34]. Finally, centres which reported promising practices were identified, with four services participating in detailed case studies around their specific practices and innovations. Based upon interviews and observations, two of these services were identified as having strong Indigenous workforces and proactive Indigenous employment strategies, and further analysis utilising the lens of Indigenous health workforce was undertaken.

## Design

Case study methodology underpins how we conducted this component of the research. Stake [35] describes case studies as "the intrinsic study of a valued particular", where the object of study is selected "not because it is representative of other cases, but because of its uniqueness, which is of genuine interest" [36]. This approach was particularly relevant for the current study, where the services were selected because they were rated to be particularly high performing and innovative in their provision of cancer services for Indigenous cancer patients and their families in the first phase of the research. If done sensitively and appropriately, it has been concluded that there is value in using case studies in Indigenous research [37, 38].

## Cultural and ethical considerations

Ethics approvals for the national study were obtained from the Western Australian Aboriginal Health Ethics Committee (WAAHEC) (approval number 483) and the Human Research Ethics Committees of University of Western Australia (RA/4/1/6286), as well as multiple other local ethics committees. Overarching ethics approval for the specific sites was obtained from the St Vincent's Human Research Ethics Committee (approval number HREC/16/SVHM/94), with site-specific approval obtained through local ethics and governance processes.

The NHMRC Guidelines for Ethical Conduct in Aboriginal and Torres Strait Islander Health Research were adhered to [39]. The values of reciprocity, respect, equality, responsibility, survival, protection, spirit and integrity, were central to this research, and provided guidance to the design and conduct. Three members of the research team are Aboriginal women. These researchers participated in the design of the study, conducted interviews, and provided input into, and approved, the draft manuscript. Prior to commencing the case studies an Aboriginal Advisory/Reference Group was formed. This group had face-to-face meetings and provided email support and advice to the study. In addition, we consulted with peak bodies of Aboriginal and Torres Strait Islander Community Controlled Health Organisations.

## Participant recruitment and profile

Recruitment was purposive with relevant staff, Indigenous cancer patients and family members identified by local health service staff and managers within each participating organisation. The site investigator ensured that all participants were informed and prepared prior to being approached by researchers for interview. Staff were invited to participate if they were involved with the care or support of Indigenous cancer patients or if they filled a leadership role in the care of Indigenous patients. Interview participation was voluntary and all participants were provided with information about the research and gave either written or oral

consent prior to data collection. Participants were reminded that they could stop the interview at any time. Indigenous patients were encouraged to include a support person or family member to be with them during the interview process.

Twenty-four in-depth interviews were conducted with relevant hospital staff (n = 24). Fourteen participants worked at the Regional Service, ten participants were employed in the Urban Service. Participants included more women (n = 20; 83%) than men, and a third (n = 8) identified as Aboriginal or Torres Strait Islander. A diverse set of professions participated, including Indigenous Liaison Officers, Social Workers, Cancer Coordinators, Registered Nurses, Oncologists, Managers, Executives, and Administration staff.

Six in-depth interviews were conducted with Indigenous patients with cancer (n = 5; four male) and affected family members (n = 3; all female). Three patients and one family member spoke about their experiences at the Urban Service, two patients and two family members described their experiences with the Regional Service.

## Data collection

Interviews were conducted between March 2015 and December 2018, mostly in-person (n = 25) but five were undertaken by telephone, and ranged in length from 30 to 90 minutes. An Indigenous researcher participated in all interviews with Indigenous patients and family members, which were all conducted in person. Interviews were conducted with the use of an interview guide, audio taped and transcribed verbatim. The interviews for the detailed service studies were conducted between September 2017 and December 2018 by four female researchers. Two of the researchers conducting interviews were Aboriginal women, both of whom have clinical backgrounds in cancer, and one of whom is an experienced researcher. The two non-Indigenous interviewers both have clinical backgrounds and over twenty years' experience with collaborative research into improving Indigenous health outcomes. We also included four staff interviews from the original service interviews conducted between March and October 2015; these were conducted via telephone by one non-Indigenous female researcher with five years' experience in Indigenous health research.

Three staff were interviewed twice, initially in 2015 as part of the original service interviews, and again in 2017 or 2018 as part of the detailed service study. The original interviews with these participants were included in the analysis as they offered rich additional insights into their service's workforce approach. An interview with one key staff member was included from the 2015 study because they were on long leave, and therefore unavailable for interview during the detailed service study. On three occasions, two staff members were interviewed together for their convenience, and two patients chose to be interviewed with a family member. Health staff were asked about initiatives and programs to improve engagement with Indigenous people, specifically within cancer services, including questions on cultural awareness programs, cultural identifiers, Indigenous staff employment strategies and numbers. Indigenous people affected by cancer were asked about their experiences at the cancer centre and any suggestions for changes or improvements.

Additional information was gathered from the health service websites, including strategic plans, Reconciliation Action Plans and information on Executive and Board membership.

## Data analysis

We used the thematic analysis process described by Green et al. [40] of immersion in the data and coding, with rereading and aggregation of identified themes. Initial coding was undertaken by an independent qualitative coder, who used NVivo 10 to help organise and extract relevant data, and identify initial themes. Discussions within the team further developed these

themes. All interviews were then re-read and manually coded by a team member (EVT) using an Indigenous health workforce lens to develop existing themes and identify additional themes. The whole team then met to further refine themes and reach agreement on the final themes. Analysis was conducted to ensure that each theme could be traced back to the original data, using direct references from the interviews to provide evidence and maintain the voice of the participants. Provisional data interpretation was checked with key stakeholders at each service, with feedback and additional information incorporated into the final analysis. All authors had input into and approved the final manuscript.

## Results

Eight themes were identified from the way that the two services supported their Indigenous workforce: strong executive leadership, a proactive employment strategy, the existence of an Indigenous Health Unit, the role of the Indigenous Liaison Officer (ILO), the inclusion of the ILO within the multidisciplinary team, the availability of professional development, a supportive work environment and a pervading culture of respect. Despite differences in rurality, management and proportion of Indigenous patients, all eight themes were identified within the data from both health services (albeit sometimes demonstrated in different ways). Hence, the elaboration of workforce themes as described was common to both services unless noted otherwise. Furthermore, participants reported two positive outcomes resulting from the active implementation or demonstration of the eight workforce themes: 'Improved Indigenous patient outcomes' and 'Improved Indigenous staff outcomes'. These themes and the reported outcomes are summarised in Fig 1 with elaboration and the supporting evidence to follow.

### Executive leadership

The Executive team in both health services demonstrated a strong and positive commitment to improving Indigenous health outcomes, strengthening the cultural safety of their health services, and to supporting and growing their Indigenous workforce. Both services have made improving Indigenous health outcomes a key commitment of their strategic plans and both have a Reconciliation Action Plan (RAP) containing multiple action items on increasing Aboriginal and Torres Strait Islander employment. The Urban Service is operated by a national not-for-profit healthcare provider, and their RAP "is the overarching strategy document to empower Australia's First Peoples" for all health care facilities that the national organisation operates. The national organisation commissioned a painting representing their RAP, which was a collaboration between three Indigenous artists and 48 staff from the three states where the health care facilities are located. The artwork is now displayed in each facility, including in multiple areas of the Urban Service, publically highlighting their commitment to the RAP. Progress on the RAP was on the agenda and discussed at every Executive meeting.

The Regional Service has Indigenous leadership at the highest levels, with an Executive Director of Aboriginal and Torres Strait Islander Health, a Board member who identifies as Indigenous, the requirement that a member of the Clinical Council must be Indigenous and an Aboriginal and Torres Strait Islander Community Advisory Council.

While the Urban Service does not have Indigenous representation on the Executive, comments from multiple participants showed that Executive support is felt by staff on the ground, epitomised by the comment "We certainly do have the support of executive members to do what we need to do to have good outcomes" (Participant 7, Manager, Indigenous). Furthermore, the Executive team of this service engages directly with the Indigenous health workforce by holding an offsite annual forum to engage with all Indigenous staff members (including support staff). The first day of the forum is attended by members of the Executive team who

**Executive Leadership**
- Commitment from Executive
- Reconciliation Action Plan and policies
- Indigenous representation
- Direct engagement with Indigenous staff

**Employment Strategy**
- Number of Indigenous staff
- Employment targets, plans and policies
- Supportive recruitment process
- Cadetships

**Indigenous Health Unit**
- Indigenous led
- Support Indigenous staff
- Defining ILO role and scope of practice
- Workload allocation

**Indigenous Patient Outcomes***
- Improved health outcomes
- Increased compliance
- Increased cultural safety
- Better supported

**Work Environment**
- Physical: flags, artwork
- Acknowledgements
- Uniforms and shirts
- Cultural events
- Cultural Awareness Training

**Indigenous Liaison Officer**
- Early, 'automatic' involvement
- Informal navigator and care coordinator
- Patient advocate

**Staff Outcomes***
- Indigenous staff feel respected
- Increased recruitment and retention of Indigenous staff
- High job satisfaction

**Professional Development**
- Career pathways
- Training opportunities and specialisation
- Mentoring and two-way learning
- Future workforce

**Multidisciplinary Team Inclusion**
- Regular meetings
- Joint assessments
- Relationships
- Input sought and referrals from clinicians

**Culture of Respect**
- *"My opinion and my judgment is respected"*
- *"I don't know how they'd go without us"*
- *"The Aboriginal Liaison community is just gold"*
- *"They are respected absolutely on their own grounds and their opinions are sought across many, many disciplines"*

* Reported in interview data

**Fig 1. Indigenous workforce support themes and reported outcomes.**

talk with and seek feedback from staff. This engagement affirms that Indigenous ways of working are recognised and valued at senior level, as captured by one participant recounting an interaction at the forum:

> *"We had a guy who started in gardens and is working in maintenance . . . We were sort of having a bit of a yarn and he said, you know, 'I see community members while I'm working and I'll say hello and I probably shouldn't' and [the CEO] said, 'No, you should. That improves the culture of the hospital as you have just identified. They feel culturally safe coming in and they can have a yarn to the maintenance guy who they know and who potentially they call uncle, but you are contributing to the cultural safety of the place, so don't stop.' So, to hear that from the high level CEO is pretty awesome."* (Participant 8, Manager, Indigenous)

## Employment strategy

Compared to most other hospitals included in the initial survey of cancer services, the two health services employ a large number of Indigenous staff members including in identified Indigenous roles. At the time of writing, the Urban Service employs 52 Indigenous staff, 0.9% of the total workforce. The Regional Service employs 241 Indigenous staff, 3.74% of the total workforce. Both services have specific and measurable targets set for the employment of Indigenous staff. The Urban Service was aiming for a state population parity target of 1% by the end of 2018, and 3% by 2022. The Regional Service has a target of 4.5% by the end of 2022, with a stretch target of 8% (equivalent to the population proportion).

In 2015, the Urban Service published an "Aboriginal and Torres Strait Islander Employment Plan". Connected to the RAP, this plan outlined objectives and an implementation framework for the whole organisation. Each section contains multiple action items with assigned responsibilities, timelines and targets. In 2019, this service implemented a cultural leave policy, an action item in the employment plan, which provides Indigenous staff with access to paid 4 days cultural and ceremonial leave. At the time of writing, the Regional Service was in the process of developing an "Aboriginal and Torres Strait Islander Workforce Strategy".

The Urban Service also demonstrates their commitment to increasing their Indigenous workforce through the existence of an identified role within the human resources (HR) department responsible for the recruitment and retention of Indigenous staff across the health service. This service regularly advertises an Expression of Interest for Indigenous staff on their careers portal, encouraging applicants for both clinical and support service positions, with the statement that any Indigenous applicants "who can demonstrate that they meet the essential position criteria will automatically proceed to interview". Furthermore, the service has a successful and rapidly growing Indigenous Cadetship Program, and a Graduate Nursing Program that supports Indigenous nursing students through the application process for Graduate positions.

## Indigenous health unit

Both health services have an Indigenous Health Unit, Indigenous-led, which is responsible for managing the ILOs, supporting all Indigenous staff, designing and delivering cultural awareness training and quality improvement. The Urban unit was only established in 2016, after the initial data collection for the cancer services study was undertaken, and at the request of the service the detailed study of the service was delayed until after the Unit was established. Importantly, both units reported adequate autonomy and independence to ensure that "what we do

is Aboriginal led" (Participant 10, Project Officer, non-Indigenous). The manager of one Indigenous Health Unit describes the management of the unit: "I have autonomy to sort of set the agenda and say, 'This is where we are going'" (Participant 7, Manager, Indigenous).

The Indigenous Health Units play an important role in managing and supporting Indigenous staff, particularly ILOs. Historically, in both services, the ILOs were based and managed on the wards where there were less opportunities to connect with other Indigenous staff members and they were unlikely to have an Indigenous manager. Both services have moved to the model of a centrally managed Indigenous team so ILOs have an Indigenous manager as well as a team of Indigenous colleagues with whom to network and debrief. A member of Executive at the Urban Service explained some of the reasoning behind this: ". . .to get more of a profile around the unit, to have their own identity. . . and not as an exclusion thing, you know. It is a support thing." (Participant 9, Executive, non-Indigenous)

Management stressed the importance of the Indigenous Health Unit in defining the ILO role and scope of practice to ensure that ILOs were not asked to work outside their role, with the aim of preventing staff from being overworked or suffering from burn out.

> *"We've also been careful that really defining the scope of practice for the Aboriginal liaison officer and making sure that they work within their scope of practice and that they are not pulled in a million different directions to do things that are not"* (Participant 7, Manager, Indigenous)

Another benefit of the centrally managed Indigenous Health Unit mentioned by staff was better allocation of workload across the team, allowing staff members to cover each other, provide back-fill for training or leave and helping to prevent staff becoming overworked.

> *"We meet every morning now. If somebody is away, 'Who will take this?' you know, 'Who will look after this ward?' and that phone is given to whoever is going to do that. So it is a shared responsibility now."* (Participant 17, ILO, Indigenous)

## Indigenous liaison officer

One factor of the ILOs' role reported by Indigenous and non-Indigenous participants was their "automatic" involvement with any Indigenous patient, usually on the patient's first visit. While reported by staff in the oncology ward, it was evident that this occurred widely at both health services, showing respect and valuing of the ILO role across these health services. In both cancer services, important system functionality had been developed which helped with both workload allocation and in ensuring that no patients were missed, with ILOs readily able to print off a list of all Indigenous patients within the service. In the Regional Service, one of the ILOs had recently instigated this improvement to the system.

Participants described many facets of the ILOs' role, with a widely expressed view that the ILOs were the "linchpin" who "joins the dots between all the players: family, patients, community, hospital staff" (Participant 10, Project Officer, non-Indigenous, 2015 interview). ILOs fulfilled an important navigator or care coordinator role as well as family liaison.

> *"When the patients come in, the Aboriginal Liaison Officers do come in with the patient. . . They do act like navigators. They bring them in. They go and book the forms and book that, book this. They kind of take them through. So they do take on a care coordination role."* (Participant 13, Oncologist, non-Indigenous)

ILOs reported feeling empowered to advocate for their patients with the doctors to ensure that patient's wishes (such as choosing to discontinue with treatment or returning home to die on country) were upheld.

*"I had another old chap, he was palliative and he wanted to go home... And it was just advocating with doctors, because they were saying, 'Oh, he has got some diabetes trouble'. Well, you know, he doesn't care about his diabetes. You know, he only just got home. We got him home and that was on the Friday ... Well, he passed on Sunday."* (Participant 17, ILO, Indigenous)

## Multidisciplinary team inclusion

ILOs were valued members of the multidisciplinary team, attending regular clinical meetings and speaking on behalf of Indigenous patients.

*"We have a Wednesday haematology meeting, and so all the patients on the ward, the ILOs would know all the patients that are Aboriginal and Torres Strait Islander patients and they would be expected to speak up for that patient, and they are very, very valued."* (Participant 20, Social Worker, non-Indigenous)

Joint patient assessments conducted by a Social Worker and an ILO were "expected". Social Workers perceived benefits such as improved cultural safety for the patient and having a respected "cultural expert" to help them advocate for the patient.

*Every time I have got someone who is Aboriginal I always have them [the ILO] introduce me as part of that cultural safety and appropriateness. So I don't just go in; I get introduced. They are there with me when I do my assessment. They bring the cultural safety component, where I do the social work assessment component, and so we work as a team with the patient, and having that cultural respect and sensitivity the whole way.* (Participant 3, Social Worker, non-Indigenous)

Staff stressed the importance of an informal, collegial relationship with their ILO colleagues, where they could "pick up the phone and chat... and know you can have that really collegial conversation about how we are going to work together to support this patient" (Participant 2, Social Worker, non-Indigenous) or "just knock on the door and you go in and talk about a case, or vice versa" (Participant 20, Social Worker, non-Indigenous). These close relationships were aided by a historical shared management structure (where the ILO team had originally been managed by the Social Work department) or close physical location, where daily informal exchanges could naturally occur.

Doctors and nurses regularly sought input from ILOs regarding Indigenous patients. As one ILO describes, "Doctors will call us while in the consult room and they will say, 'Can you come?'" (Participant 17, ILO, Indigenous). This was considered unusual, as one participant observed, "you don't see everywhere" (Participant 2, Social Worker, non-Indigenous).

The feeling of being a valued member of the team was summarised in this comment made by an Indigenous staff member:

*"But all discussions around our Indigenous patients, we're part of the discussion in those meetings. Yeah they always look to us around what's going on. And even though the social workers are involved. And referrals, yeah new referrals we will be part of. ... I mean our referrals are*

*direct from the consultants too. You know we get e-mails from them, we get phone calls from them. So we're very much a part of the team here."* (Participant 24, ICC, Indigenous)

## Professional development and two-way learning

Both services prioritised training opportunities for Indigenous staff. Career development for Indigenous staff was an action item on both RAPs, with one service aiming to develop career pathways and the other service targeting 100% of all Indigenous staff to develop a career plan with their manager as part of their annual performance plan. Indigenous staff were encouraged to consider their career pathway and were supported to achieve higher qualifications:

*"It is about growing people, not just filling a plug in a hole. So 'This is a pathway for you. Have you thought about doing nursing? Have you thought of doing allied health?' So we are constantly moving people forward. That's the goal."* (Participant 9, Executive, non-Indigenous)

*"We do have one [ILO] currently who is doing a nursing degree. She is in ED, and then there was another one. She only recently left, but she is doing her social worker degree... Yeah, they [the hospital] do encourage us."* (Participant 17, ILO, Indigenous)

ILOs were encouraged to develop a specialisation and attend training in that speciality area. One ILO in the Urban Service and three ILOs in the Regional Service worked primarily in the oncology ward, which enabled them develop expertise in oncology, build relationships with the oncology team and become familiar with the patients and their specific needs. A staff member at the Urban Service describes these benefits:

*"With [ILO], her specialisation is in renal and cancer care... so what that means is they get more embedded into teams and there is more input into developing their knowledge and skills for working in a particular area of care.... We want that worker [ILO] to have job satisfaction. We also want them to develop expertise. So today [name], who's our cancer ILO, she's off doing a two-day workshop [on cancer for ILOs]"* (Participant 10, Project Officer, non-Indigenous)

Both services described informal mentoring for ILOs. The Urban Service followed a "Working Together" model of two-way learning and experiential learning, "you have dialogue, you learn from each other" to build the capacity of both the ILOs and the non-Indigenous clinical staff:

*"I'm very, very lucky that we have our two-way learning model and that I work so closely with social work."* (Participant 4, ILO, Indigenous)

*"There is still a lot of mentoring and supervision that comes out of social work or other disciplines ... But, again, it is always two-way. It is not all that sort of top-down that the professional social worker or nurse is the teacher. It works both ways, so we continue to learn."* (Participant 10, Project Officer, non-Indigenous)

Both services actively support the development of the future Indigenous health workforce by providing clinical placements and mentoring specifically for Indigenous students studying in the fields of medicine, nursing and allied health. The Urban Service has a successful and rapidly expanding Indigenous Cadetship Program and a Graduate Nursing

Program. The Indigenous Cadetship Program is government funded and offers second and third year Indigenous nursing and allied health students paid employment beyond their clinical training. This program sits within the Indigenous Health Unit, ensuring a culturally safe entry point into the hospital program for Cadets. The Graduate Nurse Program offers a seamless transition into nursing for Indigenous nursing graduates, and involves two six-month rotations.

> *"The idea is to bring Aboriginal students in and give them exposure working in a hospital environment in order to better prepare them for the workplace... We are hopeful these cadets come back to us through the graduate nursing program and build a career here . . . We want to give them exposure so they know what to expect"* (Participant 7, Manager, Indigenous)

### Work environment

Both services have made an effort to develop a culturally safe work environment with physical representations of respect incorporated and measured in multiple action items in both services' RAP. This respect includes prominently displaying the Aboriginal and Torres Strait Islander flags throughout the hospital including at the entrance and with small flags on reception desks, artwork painted by local artists on the walls, and posters promoting Indigenous health with photos of staff members. Both services had policies specifying that an Acknowledgement of Country plaque is displayed at the entrance to the service, and that an Acknowledgement of Country is undertaken at all meetings.

Clothing was another physical expression of respect. In the Urban Service, the ILOs have their own uniform, making them easily identifiable to other staff and patients and seen as a way of legitimising the role. In the Regional Service, an Indigenous cancer patient designed a t-shirt that is worn by many staff members as a way to show support for Indigenous health:

> *"Yeah, all the radiation people have got one [T-shirt], yeah. One of our patients designed that for us. The pink part is the pink ribbon for breast cancer and the blue one is the blue ribbon for prostate cancer, and then, yeah. We particularly wear that on Friday that shirt; that is our Friday shirt"* (Participant 23, Administration Officer, non-Indigenous)

Both services hold cultural events at significant times of the year, such as Reconciliation Week and NAIDOC week (annual Australian observances celebrating the history, culture and achievements of Indigenous peoples). Action items in their RAPs encourage staff to participate in such events, and remove barriers to staff attending cultural events. A staff member at the Urban Service reported that organising cultural events was a shared responsibility and a way of building relationships between Indigenous staff and non-Indigenous staff: "It's not just ALOs doing it. There's a whole lot of staff from the hospital helping to run it. And the beauty of it is that working together experience" (Participant 10, Project Officer, non-Indigenous). In the Regional Service, an Elder who was also an ILO at the service had conducted smoking ceremonies at significant events, such as when the new palliative care centre opened.

Both services provide cultural awareness training to increase staff knowledge and understanding of Indigenous peoples' cultures, histories and achievements. An action item in the RAP of both services was to ensure that the training program is mandatory for all staff. To meet this aim, Indigenous staff at the Urban Service have developed an online training module, which has currently been completed by 84% of all staff. In addition, a Learning Coordinator employed in the Indigenous Health Unit is responsible for delivering face-to-face cultural awareness training.

## Culture of respect

The deep respect felt for Indigenous staff at these services came through strongly in every interview. Indigenous staff reported feeling respected and supported: "My opinion and my judgment is respected and what I am suggesting about how we move forward is considered" (Participant 7, Manager, Indigenous). "I don't know how they'd go without us really" (Participant 24, Cancer Coordinator, Indigenous).

This deep respect felt by non-Indigenous staff members towards Indigenous colleagues was expressed at all levels of the organisation. A member of the Executive team made this observation about an ILO: "She doesn't want to be any trouble, but I mean I would walk on hot coals for her" (Participant 9, Executive, non-Indigenous). An oncologist described the ILOs as "just gold" (Participant 14, Oncologist, non-Indigenous). Recurring comments reiterated this esteem: how lucky they were to have their Indigenous staff members, how much respect they felt for their Indigenous colleagues and that the opinions of their Indigenous colleagues were sought and valued. "They are respected absolutely on their own grounds and their opinions are sought across many, many disciplines" (Participant 2, Social Worker, non-Indigenous).

## Outcomes

Participants described multiple advantages to having a strong Indigenous workforce within their health service with both improved Indigenous patient outcomes and improved staff outcomes.

**Indigenous patient outcomes.** Multiple participants reported that having Indigenous staff members, particularly ILOs, involved in patient care improved outcomes for Indigenous patients. Participants shared stories of patients achieving better health outcomes through the involvement of Indigenous staff, their increased treatment adherence, and how Indigenous patients felt more culturally safe and were better supported. A Social Worker at the Regional Service described how an Indigenous patient would have a better outcome with his chemotherapy because of the involvement of the ILO:

*"There are some times that if they [the ILO] weren't involved we wouldn't have had the outcomes. Like, for example, there was a young man from the remote area of [locality]... He was having lots of treatment, and it was very difficult getting him to appointments. And then he had to go to rehab, and this was a very big decision for him that he wanted to get into rehab. But just physically getting him there and, you know, not having a support person here, it was only the fact that the ILO went out and spoke to him and sorted some stuff out that he has actually now has gone to rehab. Now that he is doing that he is going to have better outcomes for his treatment when he has his chemo and things like that."* (Participant 20, Social Worker, non-Indigenous)

Managers and patients at the Urban Service observed that ILOs increased treatment compliance and reduced discharge against medical advice (DAMA).

*"The people that I liaise with, whether it is nurse unit managers or nurses, will always give good feedback... 'Thanks to the efforts of [ILO]' say 'we were able to get the patient to come back in' you know, in a situation where someone might be deemed as absconded or something like that, that we can do that, to be able to get the patient back in and back into the bed. So the feedback has been good."* (Participant 7, Manager, Indigenous)

*I have really appreciated the support I've received from the Aboriginal liaison officer... they have gone above and beyond what they have to do just to make sure I'm right. Without their*

*support I don't know where and what headspace I would be in. I probably wouldn't even still be in hospital. They have done all they can to keep me here and, yeah, I probably would have done a runner and gone back to [town] by now if it wasn't for their support and understanding.* (Patient 3)

Indigenous staff increased the cultural safety of the service, not only through their presence and interactions with patients, but also through advising non-Indigenous staff.

*"They send different people to you like oncology psychologists and things like that, but I didn't really want to open up to them, do you know what I mean? But, as I said before, once again I had [names staff in the Indigenous Health Unit], another bloke over at the Aboriginal liaison, so they were handy to talk to."* (Patient 3)

*"Aboriginal liaison officers are fantastic in that regard, you know. . .. before the consultation and after the consultation in advising us how to do this now, how to go about this with this particular family, where the balance of power lies with this family. . . The patients see it as an acknowledgement of the fact that this is not just a medical consultation. . . There is a cultural acknowledgement."* (Participant 14, Oncologist, non-Indigenous)

Staff and patients from both services reported that having ILOs resulted in the patients receiving more "holistic" support. ILOs sorted out "the day-to-day things that, you know, sometimes we don't think of, but that are important to [the patient]" as well as "all that sort of logistical stuff" (Participant 19, Nurse, non-Indigenous). A patient at the Regional Service recalled how the ILO had helped when they first arrived: "Yeah, [the ILO] helped us out a lot here too. . .. they helped me out with food vouchers and all that sort of stuff when I first came down because we had no money, you know." (Patient 4)

**Staff outcomes.**   The Indigenous staff interviewed spoke positively about working for the health services. Staff members reported feeling "respected", "listened to" and "part of the team". One Indigenous staff member reported choosing to work at the health service because of its positive reputation within the Indigenous community:

*"Some of the reason I did want to come here [work in this hospital] is the community trust the organisation in the hospital. So, for me, I previously worked at [hospital name] hospital and there is probably a little bit of mistrust of the Aboriginal community at that hospital . . . So to come to a hospital that the community actually put their hand up to go to was an interesting sort of thing for me . . ."* (Participant 8, Manager, Indigenous)

Both services were proactive and doing well with recruitment and retention, with a number of long-term Indigenous employees at both services. At the time of writing, neither service had any unfilled vacancies for identified roles. The Urban Service's Cadetship Program has supported over 20 Indigenous students through to the completion of their degrees and into graduate positions since its inception in 2012. Although there was no guarantee these students would remain with the health service after graduation, they increased the number of Indigenous staff working at the hospital. As one staff member pointed out, increased employment of Indigenous staff became a 'virtuous cycle' with those increased numbers leading to better support and a safer work environment:

*"The more Aboriginal people work here, the more career structure they have, the more opportunities they have, the more safe an environment it becomes, and the more the people who work with Aboriginal colleagues learn about how to be respectful and responsive and a lot of*

*stuff happens by osmosis, not just through cultural awareness training."* (Participant 10, Project Officer, non-Indigenous, 2015 interview)

Indigenous staff members reported feelings of job satisfaction resulting from being able to improve outcomes for the community through their work at the health service.

*"So I feel that my role here is to try and make our mob—and when I say 'our mob' I am talking about Aboriginal and Torres Strait Islander people—I am trying to make it more culturally safe for them. . . so that when they go home, if they come again and they go, 'No, it will be okay, I'm looked after'".* (Participant 16, ILO, Indigenous)

Non-Indigenous staff also recognised the benefits they received from working closely with their Indigenous colleagues, not only in patient outcomes and team moral, but also in improving their understanding of Australia's First Peoples and their culture. They spoke of the personal satisfaction they felt to be working for organisations that prioritise Indigenous health and care for both Indigenous staff and patients.

## Gaps and continuous improvement

This study is cross sectional and reports on the services, their programs and progress, at a particular point in time. Both services were aware of areas where they wanted to improve and were actively and continually trying to improve Indigenous health outcomes, strengthen the cultural safety of their health services, and support and grow their Indigenous workforce. Both services demonstrated this commitment through new policies and procedures that were under development at the time of writing, such a new RAP (Urban Service) and an "Aboriginal and Torres Strait Islander Workforce Strategy" (Regional Service). In addition, the Urban Service employed a member of the Indigenous Health Unit whose role included a focus on continuous improvement for Indigenous health.

Despite the efforts of both services, gaps in the areas of recruitment and orientation were discernible from the interview data. One participant spoke at length about the recruitment process at the Urban Service. He described how the process and position requirements were discouraging and led to Indigenous people not applying (because they believed they weren't eligible) or eligible candidates not being considered for a role because they didn't hold a required qualification (even though their previous work experience rendered them eligible). However, a chance discussion with a colleague from HR led to the General Manager of HR seeking feedback from this participant about their experience with the recruitment process:

*"I get this email from [the head of HR] saying. . . 'What were the issues for you as an employee applying for a job at [health service]?' And I said, 'Well, you know, the process.'. . . 'How hard was it to fill out the application?' and I said, 'It was a nightmare. You know, the first thing I looked at was you need a degree and I ain't got that, so there's no point in me putting in for it.'"* (Participant 8, Manager, Indigenous)

Action items in the Urban Service's "Aboriginal and Torres Strait Islander Employment Plan" and the Regional Service's RAP show that the services are aware of, and working to address this gap. Targeted action items include: "assess current recruitment processes and identify barriers for Aboriginal and Torres Strait Islander job-seekers", "offer all Aboriginal and Torres Strait Islander applicants the option of including an Aboriginal and/or Torres Strait Islander panel member on selection panel" and "interviews to be held in a relaxed

environment that includes cultural recognition, Aboriginal and Torres Strait Islander flags and artwork".

Orientation programs for new employees was another area where the data suggested opportunities for improvement. Indigenous staff members at both services talked about how "overwhelming" and confusing it was to start a new role with minimal orientation, training or handover. One staff member did not realise that she was only employed on a six-month contract when she started with the service (although her role had since developed into a permanent position). When asked about orientation, this participant laughed and responded, "I started on December 17. . . So I'm here for not even a week, I think, and the other liaison officers go on leave. And so I'm here by myself and 'Shoot! What the hell!'" (Participant 4, ILO, Indigenous). Another participant described the challenge of joining the service to undertake the newly created role of Indigenous Health Unit Manager:

> "Yes, it was a little daunting because the tricky part is you don't have a predecessor to sort of -- You know, there are no handover notes or anything like that. It was a little bit of a choose-your-own-adventure to begin with but you are also coming into a hospital environment where you've got more responsibility in your role. It was tricky having to learn the business and then also get an idea of what was going on with my job." (Participant 7, Manager, Indigenous)

Action items in the Urban Service's "Aboriginal and Torres Strait Islander Employment Plan" such as "include Aboriginal and Torres Strait Islander employment and cultural safety to general orientation" and "extended orientation program for Aboriginal and Torres Strait Islander employees to [health service] provided as required" show that the service is working to address this gap. The Regional Service's RAP has an action item to "develop strategies for supervision and mentoring for new and existing Aboriginal and Torres Strait Islander staff".

## Discussion

Despite considerable rhetoric and goodwill around closing the gap, and the need for Aboriginal health workforce and cultural security in care delivery, many health services fall well short when it comes to evidence of achieving progress. It is clear from examining the two cancer services and their affiliated hospitals described in this paper that support for their Indigenous workforce underpins their achievements and is reflected in their strategies, key policies and initiatives. Despite evident differences between the services, both of them fitted our criteria of providing innovative services in cancer care for Indigenous patients, and it seems unlikely this could occur without them having a strong Indigenous workforce and proactive Indigenous employment strategies. The data makes clear that a supportive workplace for Indigenous staff is created through the actions of the health service as a whole. We identified the following elements: strong executive leadership, proactive policies and employment strategies, supportive work environment and respectful culture; the existence of an Indigenous Health Unit to ensure that Indigenous staff (particularly ILOs) are appropriately managed and supported; and implementation of the policies and strategies within the specific hospital departments studied, ensuring that Indigenous staff are included, supported and developed in those departments. The benefits of these strategies were demonstrated in the positive patient and staff outcomes described by the participants in this study. While Indigenous staff reported that recruitment and orientation were areas where there were gaps when they were appointed and still required attention, efforts made by the services to continuously improve their support for the Indigenous workforce indicate that some of these areas had already been improved.

The strong leadership and policies of the health service as a whole were found to be vital for the support of the Indigenous workforce. Through our case studies with cancer services around Australia, we observed that it "starts from the top"; if the commitment to Indigenous health and workforce is dependent on just one or two dedicated employees then efforts can diminish over time as supportive staff leave or get burnt out. This reflects findings by other studies, including Jongen et al. [41] who found that strong leadership was essential for a strong and stable Indigenous primary health care workforce and the Mason Review of Health Workforce Programs, which states "Aboriginal and Torres Strait Islander leadership is recognised internationally as a key factor in the development and sustainability of programs aimed at increasing Aboriginal and Torres Strait Islander workforce capacity, and influencing the non-Aboriginal and Torres Strait Islander health workforce to provide culturally safe and appropriate services" [42].

For real progress in building an Indigenous health workforce in a mainstream health service, the commitment to Indigenous health needs to be front and centre in the service's values or mission statement, as occurred with the two services in this study, and has been reported previously [43]. This complies with Action 1.2 in the Australian National Safety and Quality Health Service (NSQHS) Standards, which states that the health service governing body should ensure "that the organisation's safety and quality priorities address the specific health needs of Aboriginal and Torres Strait Islander people" [44]. In these services the commitment to build a strong Indigenous workforce followed on from a long-term and ongoing commitment to Indigenous health. This commitment was given expression in an active RAP which made accountable commitments to workforce support, particularly evident in the Urban Service through an Indigenous Employment Strategy with specific and measurable targets set for the employment of Indigenous staff. Government policies and funding can provide an important driver for organisations to make improvements in this space [34, 45, 46]. In the case of the Urban Service, the state government had made available funding a few years earlier to organisations that developed an Indigenous employment strategy, so receiving this funding provided extra impetus and important resources for the health service to develop their strategy. It needs to be noted however, than organisations needs to be alert to such opportunities, and have personnel willing to seek out and apply for funding.

Work environment is a fundamental predictor of retention for the Indigenous health workforce [30, 32, 41]; however, a culturally safe work environment does not just happen through good will and good intentions. Both services have made consistent and ongoing efforts over a number of years to develop a culturally safe work environment, expressed both in the physical environment (such as flags, artwork and Acknowledgement of Country plaques) and the culture of the organisation, which is vital for creating a culturally safe space for Indigenous patients [21, 32, 47]. The importance of the physical environment in creating a culturally safe environment has been previously reported [16, 34]. Action 1.33 in the NSQHS Standards states that services should demonstrate "a welcoming environment that recognises the importance of the cultural beliefs and practices of Aboriginal and Torres Strait Islander people" [44], the importance of this is emphasised as one of six specific actions to meet the needs of Indigenous Australians in the NSQHS Standards User guide for Aboriginal and Torres Strait Islander health [48].

Developing the culture of an organisation, while less tangible than the physical environment, is both harder and of greater importance for creating a culturally safe workplace [30, 32]. Both services worked to develop the culture of the organisation, including through mandatory cultural awareness training. Having "strategies to improve the cultural awareness and cultural competency of the workforce" to meet the needs of Indigenous patients is an Action in the NSQHS Standards [44]. It is supported by a multitude of recommendations in the

literature that health staff at all levels receive mandatory ongoing cultural safety training, with completion embedded into performance management and/or professional development requirements [45, 49, 50]. Improving the cultural competency of non-Indigenous staff at all levels helps to create more culturally sensitive workplace environments, and improve respect for and understanding of Indigenous health professionals [48, 50–52]. However, mandatory cultural awareness-training programs alone cannot explain the pervading culture of respect for Indigenous staff observed by researchers at both health services. Many health services have mandatory cultural training, yet racism and stigma are major barriers faced by Indigenous health professionals throughout Australia [52–54]. What sets these health services apart is the ethos that "what we do [in Indigenous health] is Aboriginal led", combined with a commitment to what the Urban Service described as a "Working Together" model of two-way learning and experiential learning.

The way that Indigenous staff (particularly ILOs and Indigenous Health Workers/Practitioners [IHWs]) are managed and how their role is defined is linked to retention [30, 32, 41]. There are multiple reports in the literature of Indigenous health professionals experiencing lack of support from management, lack of cultural supervision, isolation and role ambiguity, all of which lead to low levels of job satisfaction, negative perceptions of the work environment, emotional exhaustion and high turnover [20, 50, 54–56]. In contrast, both health services in this study have created a centrally managed Indigenous-led Indigenous Health Unit, which is responsible for managing the ILOs and supporting all Indigenous staff, and enables the team of Indigenous staff to network with colleagues both Indigenous and non-Indigenous. In both services, the Indigenous Health Unit took responsibility for defining the ILO role and scope of practice to ensure that ILOs were not asked to work outside their role, as well as in managing workload allocation, to ensure that staff were not being overworked or at risk of burn out. The literature makes clear that having clearly documented roles, scope of practice and responsibilities helps empower Indigenous health professionals in their roles. This occurs through managing their workloads, clarifying employee, employer and community expectations, and encouraging productive working relationships, which in turn contribute positively to retention [30, 32, 57].

Although the importance of ILOs came through strongly in the interviews, there was very little mention of Indigenous staff in other clinical roles. The Indigenous staff interviewed were employed in a variety of roles (three ILOs, two in management positions, two nurses and one in administration), however references to Indigenous staff in other roles (such as Indigenous nurses, social workers and a physiotherapist) were brief and incidental. There was no mention of Indigenous doctors. This is despite the known importance of Indigenous people filling a variety of clinical roles, and the almost two-decade long push to increase the number of Indigenous doctors [49, 58, 59]. There are a number of possible reasons for this. Firstly, ILOs are in identified roles, which makes them more evident within the organisation than Indigenous colleagues in nursing, medicine and allied health, who may prioritise identifying in their health professional role over that of being Indigenous. As stated in the introduction, Indigenous Australians are under-represented in the health workforce, with large disparities between rates of Indigenous and non-Indigenous employees for every health profession, including nurses (in 2015, 1.1% of nurses and midwives identified as Indigenous) and doctors (0.5% of doctors identified as Indigenous) [60, 61]. Furthermore, the focus of this study was on cancer services, with specific questions about the ILOs who worked in oncology. If we had interviewed additional staff from other areas of the health service, we may have collected more data on Indigenous people in other clinical roles.

Although strong health service leadership, mission statements and policies are vital for the support of the Indigenous workforce, there are numerous examples in the literature of

managerial directives and service-wide policies and procedures not being translated into practice "on the ground" [13, 62, 63]. However, it was these cancer services' respect for and support of their Indigenous staff that led us to examine the affiliated health services more broadly. The genuine valuing of Indigenous staff came from the deep respect in seeing the advantages that they brought to the patient care role and was manifest in their automatic inclusion in patient care team meetings, not at observers but as consultants to senior health professional staff. This contrasts with the literature, where although there have been calls both in Australia and internationally to create a more diverse cancer workforce capable of providing culturally safe cancer care [64–66], we found few reports of cancer services successfully supporting or developing an effective Indigenous health workforce. Successful initiatives included two Indigenous patient navigator (IPN) programmes in the USA and a pilot IPN programme in Australia [67, 68] and an Indigenous palliative care workforce education program in Australia [69].

## Recommendations

Health services have an important role to play in supporting and growing the Indigenous health workforce. The strategies and actions captured in Fig 1 can be used by health services and cancer services to inform their own policies, programs and resourcing to support building their own Indigenous workforce. Where a recommendation below aligns with an Action in the Australian NSQHS Standards, the action number is given in brackets. The NSQHS Standards User guide for Aboriginal and Torres Strait Islander health suggests additional strategies for health service organisations seeking to improve the quality of care and health outcomes for Indigenous people and support their Indigenous health workforce [48]

Health Service Recommendations

• Demonstrate commitment to improving Indigenous health outcomes and achieving Indigenous health workforce equity via the development and proactive support of a mission/vision statement and appropriate policies and processes, including a Reconciliation Action Plan. (NSQHS Actions 1.2 and 1.4 [44])

• Develop knowledge and understanding of the current Indigenous workforce to inform future employment and professional development opportunities. (NSQHS Action 1.4 [44])

• Increase engagement with Indigenous staff, patients and communities and seek input on Indigenous health, system function and workforce development activities. (NSQHS Action 2.13 [44])

• Create a welcoming physical environment through the display of Acknowledgement of Country plaques at building entrances, Aboriginal and Torres Strait Islander flags, and local Indigenous artwork in all departments. (NSQHS Action 1.33 [44])

• Assess current recruitment processes and remove barriers for Indigenous job-seekers.

• Develop a culturally considered welcome to the organisation, such as including cultural safety expectations in the general orientation.

• Create a supportive environment for Indigenous employees through initiatives such as establishing an Indigenous employee network or providing access to culturally safe mentoring relationships.

• Encourage all Indigenous staff to develop and action a career plan with their manager and develop appropriate career pathways for Indigenous staff.

- Ensure all staff complete mandatory health-focused cultural awareness training, with completion embedded into performance management and/or professional development requirements. (NSQHS Action 1.21 [44])

- Hold and participate in cultural events at significant times of the year (such as NAIDOC Week and Reconciliation Week), with encouragement for all staff to participate in these events.

- Create ongoing Indigenous traineeship positions and cadetships.

- Incorporate data collection, analysis and evaluation into new strategies and programs with the publication of results where possible.

- Take time to listen to Indigenous staff around their thoughts on potential service improvements and work to implement their recommendations

   Cancer Service Recommendations

- Implement systems to ensure that ILOs and IHWs are notified of new Indigenous cancer patients.

- Invite ILOs and IHWs to multidisciplinary meetings and give them the opportunity to provide input on their patients.

- Provide ILOs and IHWs with professional development opportunities such as in-house workshops and mentoring, external training and conference attendance.

- Ensure that staff follow the two-way learning model to build the capacity of both the Indigenous and non-Indigenous staff through opportunities for mentoring and working together.

Government also has a role in delivering and supporting effective schemes that provides incentives and financial support for health services to develop Reconciliation Action Plans, Indigenous employment policies and Indigenous cadetships and graduate programs.

## Limitations

The original intention was for this study to utilise a mixed methods approach, and to incorporate quantitative data such as patient pathway timeframes to support the qualitative data. However, there were a number of delays with this project as well as challenges with approval processes and timeframes, which led to reduced capacity and meant the quantitative component of the research was not progressed.

We interviewed only eight Indigenous people affected by cancer, with several factors contributing to this. These are vulnerable people, so participants were only approached if it was felt their state of mental health and physical well-being was adequate; this limited the number of participants approached. A number of Indigenous patients agreed to be interviewed, however, on the day of the scheduled interview they were unavailable due to poor health or recent discharge.

## Conclusion

These two cancer services and their affiliated health services may appear similar to many other health services and cancer services around the country, but deeper examination reveals why they were identified as innovative services in terms of their approach to Indigenous cancer care in Australia. These services aren't using their location, their management structure, their patient cohort, the availability of Indigenous staff or lack of funding as an excuse for inaction;

instead they have committed to improving Indigenous health and worked steadily to improve health care delivery. Importantly, rather than tokenistic inclusion of Indigenous staff in their health services, they have committed to growing the Indigenous workforce, adopting a culturally secure model of workforce support and continue to actively seek and embrace opportunities to develop further in this area. These two very different health services show how positive patient outcomes and a strong Indigenous health workforce can be achieved when a health service has strong leadership, commits to an inclusive and enabling culture, enables two-way learning and develops specific support structures appropriate for Indigenous staff.

## Acknowledgments

We thank the study participants who generously shared their views and experiences. We thank Fay Halatanu and Amy Brown for their assistance in coordinating interviews and for their input into the study; Michele Holloway for analysing interview data and providing input into the study; Carolyn Lethborg for her assistance with the study; Margaret Haigh for her input into the conceptualization of the study and for organising ethics approvals; and Nicole Rawson for assisting with data coding using the NVivo 10 software program.

## Author Contributions

**Conceptualization:** Marilyn Lyford, Toni Mason, Sabe Sabesan, Sandra C. Thompson.

**Data curation:** Emma V. Taylor, Marilyn Lyford.

**Formal analysis:** Emma V. Taylor.

**Investigation:** Emma V. Taylor, Marilyn Lyford, Lorraine Parsons, Sandra C. Thompson.

**Methodology:** Marilyn Lyford, Toni Mason, Sabe Sabesan, Sandra C. Thompson.

**Supervision:** Sandra C. Thompson.

**Writing – original draft:** Emma V. Taylor.

**Writing – review & editing:** Emma V. Taylor, Marilyn Lyford, Lorraine Parsons, Toni Mason, Sabe Sabesan, Sandra C. Thompson.

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
