## [Decision Letter · Decision Letter 0]

23 Jun 2020

PONE-D-20-06487

"We’re very much part of the team here":  A culture of respect for Indigenous health workforce transforms Indigenous health care

PLOS ONE

Dear Dr. Taylor,

Thank you for submitting your manuscript to PLOS ONE. After careful consideration, we feel that it has merit but does not fully meet PLOS ONE’s publication criteria as it currently stands. Therefore, we invite you to submit a revised version of the manuscript that addresses the points raised during the review process.

We look forward to receiving your revised manuscript.

Kind regards,

Melissa T. Baysari

Academic Editor

PLOS ONE

Additional Editor Comments: 

Please carefully consider the reviewers' comments. In addition, I note the manuscript is very long, particularly the results section. I appreciate that this is a qualitative paper, but please consider if any content can be summarised or shortened.

Journal Requirements:

2. When reporting the results of qualitative research, we suggest consulting the COREQ guidelines: http://intqhc.oxfordjournals.org/content/19/6/349. In this case, please consider including more information on the number of interviewers, their training and characteristics; how sample size was determined; and whether bias issues were considered.

4. Please amend the manuscript submission data (via Edit Submission) to include authors Marilyn Lyford, Lorraine Parsons, Toni Mason, Sabe Sabesan, and Sandra C. Thompson.

Reviewers' comments:

Reviewer's Responses to Questions

**Comments to the Author**

1. Is the manuscript technically sound, and do the data support the conclusions?

Reviewer #1: Partly

Reviewer #2: Yes

2. Has the statistical analysis been performed appropriately and rigorously? 

Reviewer #1: No

Reviewer #2: Yes

3. Have the authors made all data underlying the findings in their manuscript fully available?

Reviewer #1: Yes

Reviewer #2: Yes

4. Is the manuscript presented in an intelligible fashion and written in standard English?

Reviewer #1: Yes

Reviewer #2: Yes

5. Review Comments to the Author

Reviewer #1: Your research needs to embody the true principals of Indigenous research methodologies. There is no information on how we Aboriginal people were involved in the process. ALso, why did it take so long to collect data?

Reviewer #2: re. We’re very much part of the team here": A culture of respect for Indigenous health

workforce transforms Indigenous health care

The authors have presented a well articulated and presented paper defining the key role of workforce to support improved health for people receiving cancer care, which promotes the qualitative experiences of an Indigenous Health Workforce, and non-Indigenous colleagues, and explores the operating environments of the workforce within the wider health service.

This is an important and current topic, to improve the care experiences within mainstream health services, and reports on the outcomes of significant service investments to embed cultural safety and respect. I congratulate the health services who were profiled, and the research team who express this work.

I have minor comments and revisions.

Abstract Intro: sentence 1 is negative framing. Perhaps swap the first 2 sentences of the abstract. I note the Intro first sentence is strength based, and the remainder of the paper is strengths based.

Method

I note a responsive and flexible approach to qual interviews was undertaken

The approach to defining qualitative themes and data interpretation was appropriate, and further enhanced by confirmation of key findings within the health service

I note a careful construction to the basis of enabling culturally safe workplaces, which are then able to support enhanced care opportunities and care outcomes. I note in these health services, that culturally safe workplaces resulted from a whole of service approach to workforce - exec, human resources, targets and stretch targets, policy, and activities, and governance. The paper showed that performative activities aligned to RAP's from the 2 health services were not isolated from, but enhanced respect for Indigenous culture.

Many professional and service roles were described, with a large emphasis on ILO, yet there was no reference to the importance of Indigenous people in medical pathways (as junior doctors, general practitioners and consultant oncologists and radiation oncologists or allied health. If these were not identified in the thematic analyses, perhaps this can be explored in the discussion.

Discussion

1. I note absence of discussion against other publications in Australia or overseas on Cancer care workforce - this would help to see if these local strategies were aligned to best practices elsewhere, or indeed are likely to be internationally leading innovations.

2. Many recommendations for health services are consistent with Australian Commission on Safety and Quality of Health Care, including care for Aboriginal and Torres Strait Islander peoples. I would recommend these publications be cited. I note the appropriate citation of strategic papers for Australian Indigenous workforce.

3. Please address the comment above about Indigenous peoples within the medical workforce

4. Strengths and Limitations of methods- I suspect there was saturation of themes within the qual interviews- can this be confirmed. Do the authors regard this methodological approach as wholly satisfactory, or would they suggest modification, if this was extended to more sites in the future who may still be on a workforce innovation trajectory?

Conclusion

Strengths-based, action focussed. Appropriate conclusion.

6. PLOS authors have the option to publish the peer review history of their article (what does this mean?). If published, this will include your full peer review and any attached files.

Reviewer #1: No

Reviewer #2: Yes: Associate Professor Jaquelyne T Hughes

---

## [Author Response · Author response to Decision Letter 0]

12 Aug 2020

Thank you for allowing us to revise the manuscript according to the academic editor and the reviewers’ comments. 

We thank the editor and the reviewers for their comments on the article. Please find our point-by-point response to academic editor and the reviewers’ comments below. 

Academic Editor’s Comments 

1. The manuscript is very long, particularly the results section. I appreciate that this is a qualitative paper, but please consider if any content can be summarised or shortened. 

1: We have gone through the manuscript and attempted to shorten/summarise the content where possible. We have also reduced the number of quotes given in the results section. However, while tightening up the manuscript, in responding to the Journal Requirements and the detail requested by the Reviewers’ comments, the overall length of the paper has not been reduced. 

---

Journal Requirements 

1. Ensure that your manuscript meets PLOS ONE's style requirements, including those for file naming. 

1: Done, with minor alterations made. We believe the manuscript meets PLOS ONE’s style requirements, including those for file naming. 

2. When reporting the results of qualitative research, we suggest consulting the COREQ guidelines. In this case, please consider including more information on the number of interviewers, their training and characteristics; how sample size was determined; and whether bias issues were considered. 

2: We have provided the following additional information on the number of interviewers, their training and characteristics: 

The interviews for the detailed service studies were conducted between September 2017 and December 2018 by four female researchers. Two of the researchers conducting interviews were Aboriginal women, both of whom have clinical backgrounds in cancer, and one of whom is an experienced researcher. The two non-Indigenous interviewers both have clinical backgrounds and over twenty years’ experience with collaborative research into improving Indigenous health outcomes. We also included four staff interviews from the original service interviews conducted between March and October 2015; these were conducted via telephone by one non-Indigenous female researcher with five years’ experience in Indigenous health research. 

We have provided the following additional information (in italics) on sample size: 

Recruitment was purposive with relevant staff, Indigenous cancer patients and family members identified by local health service staff and managers within each participating organisation. Staff were invited to participate if they were involved with the care or support of Indigenous cancer patients or if they filled a leadership role in the care of Indigenous patients. 

And in the new Limitations section: 

We interviewed only eight Indigenous people affected by cancer, with several factors contributing to this. These are vulnerable people, so participants were only approached if it was felt their state of mental health and physical well-being was adequate; this limited the number of participants approached. A number of Indigenous patients agreed to be interviewed, however, on the day of the scheduled interview they were unavailable due to poor health or recent discharge.

3. We note that you have indicated that data from this study are available upon request. PLOS only allows data to be available upon request if there are legal or ethical restrictions on sharing data publicly. 

3: The qualitative dataset analysed during the current study cannot be shared publicly due to the risk of compromising the confidentiality of participants. 

It is difficult to maintain true anonymity of the services when two of the co-authors work for the services, the name and location of which must be stated in their author affiliation. Therefore, the transcripts cannot be shared, because even de-identified, there is the risk that staff and patients could be identified from the information within the transcripts. This would violate the following agreement to which the participants consented: 

Your answers will be stored on a computer but your name will not be connected with what you have said. No one will be able to match your name to the answers. All the interview information will be safely kept in a locked cabinet and only the primary researchers will be accessing this information. We will never use your name in any of the reports and publications about the research. We will present so that no-one will be able to identify you unless we have your written consent.

Limited de-identified data is available from the WA Centre for Rural Health on reasonable request.

+61 08 9956 0200

admin-wacrh@uwa.edu.au

4. Please amend the manuscript submission data (via Edit Submission) to include authors Marilyn Lyford, Lorraine Parsons, Toni Mason, Sabe Sabesan, and Sandra C. Thompson. 

4: Done.

---

Reviewers’ comments 

Reviewer 1:

1. Introduction: it would be good to put more context on the profile of Aboriginal peoples then connect this to cancer. Cancer is higher in our communities, however there needs info on the factors, including risk factors, that lead to this. 

1: We have added a sentence to provide more context on the profile of Aboriginal peoples (indicated below in italics): 

Aboriginal and Torres Strait Islander peoples are strong and resilient, with richly diverse and complex cultures which have developed over 50,000 years. Representing 3.3% of the total Australian population, Aboriginal and Torres Strait Islander peoples reside across all Australian states and territories [1]. However, colonisation had a devastating impact on traditional lifestyles, while systemic discrimination and disadvantage means that Australia’s First Peoples continue to experience lower levels of education and employment, poorer health and shorter life expectancy [2-4]. 

We already had two lengthy sentences at the start of the second paragraph of the Introduction on the factors that contribute to a higher incidence of cancer in Aboriginal peoples (excerpt below). We had not provided more information on the risk factors because the purpose of this paper was to focus on cancer treatment, rather than causes or prevention. However, we have added a sentence (indicated below in italics) on the behavioural risk factors of cancer. 

A number of factors contribute to the disproportionate levels of poor health experienced by Indigenous Australians, including socioeconomic disadvantage; lack of culturally appropriate health services and lower access to health services; and biomedical, behavioural, and environmental factors [1, 8, 9]. Indigenous Australians are more likely to have lifestyle and behavioural risk factors for cancer such as smoking, poor nutrition and physical inactivity [10]. However, reasons for poorer outcomes for Indigenous cancer patients are multifactorial and include lower participation rates in screening programs; later stage at diagnosis; the presence of comorbidities; lower uptake and completion of cancer treatment; and entrenched racism in the health system [7, 11-13]. 

2. Methodology: as an Aboriginal scholar, I would like to know this research embodied the principals and practices of Indigenous Research methodology and Indigenous standpoint theory. The theory lacks any theoretical knowledge on methodology. 

2: We have added the following information on the research design and methodology, as well as an additional section in Methods titled ‘Cultural and ethical considerations’. 

Case study methodology underpins how we conducted this component of the research. Stake [34] describes case studies as “the intrinsic study of a valued particular”, where the object of study is selected “not because it is representative of other cases, but because of its uniqueness, which is of genuine interest” [35]. This approach was particularly relevant for the current study, where the services were selected because they were rated to be particularly high performing and innovative in their provision of cancer services for Indigenous cancer patients and their families in the first phase of the research. If done sensitively and appropriately, it has been concluded that there is value in using case studies in Indigenous research [36, 37].

The NHMRC Guidelines for Ethical Conduct in Aboriginal and Torres Strait Islander Health Research were adhered to [38]. The values of reciprocity, respect, equality, responsibility, survival, protection, spirit and integrity, were central to this research, and provided guidance to the design and conduct.

3. How did Aboriginal people have governance of the study? 

3: The research project was a component of research within the Centre for Research Excellence (CRE), Discovering Indigenous Strategies to improve Cancer Outcomes Via Engagement, Research Translation and Training Centre of Research Excellence (DISCOVER-TT). The CRE was led by an Indigenous researcher, and involved Indigenous and non-Indigenous people working together to improve services and outcomes for Indigenous people with cancer. This study was done with the knowledge and support of the lead Indigenous researcher within the CRE. The CRE had its own Indigenous Advisory Group.

For the case study component of this research, an Aboriginal Advisory/Reference Group was formed. This group had face-to-face meetings and provided email support and advice. 

We consulted with peak bodies of Aboriginal and Torres Strait Islander Community Controlled Health Organisations such as VACCHO (Victorian Aboriginal Community Controlled Health Organisation) and QAIHC (Queensland Aboriginal and Islander Health Council), as well as ATSIHLAC (the Townsville Health Service Aboriginal and Torres Strait islander Health Leadership Advisory Council). 

We have incorporated this information into the paper at various points through the Methods section. 

4. Was the research team filled with non-indigenous researchers? If so, how did they exercise cultural safety? 

4: Three members of the research team are Aboriginal women, two are also co-authors on this paper. These researchers participated in the design of the study, conducted interviews, and provided input into, and approved, the draft manuscript. An Aboriginal researcher participated in all interviews with Indigenous patients and family members. We have provided additional information about the research team in the paper to clarify this point. 

The non-Indigenous members of the research team have all completed cultural awareness and sensitivity training prior to joining the research project. 

5. How were member checks done? 

5: Participants were informed prior to their interview that they could request to read and amend their interview transcripts. 

Provisional data interpretation was checked with key stakeholders at each service, with feedback and additional information incorporated into the final analysis.

6. How were they involved in the analysis and interpretation of the data? 

6: Initial coding was undertaken by a non-Indigenous, independent qualitative coder, with further analysis and coding performed by non-Indigenous members of the research team. All members of the research team, including the Aboriginal researchers, discussed the themes and reached agreement on the final themes. All authors had input into the final manuscript.

7. Given they interviewed cancer patients and staff, how did they manage GL&T associated with cancer and dying? 

7: Aboriginal patients were encouraged to include a support person or family member to be with them during the interview process. 

All participants were reminded that they could stop the interview at any time. 

In the event of a participant (staff, patient of family member) showing distress or discomfort during the course of an interview, the interviewer suspended the interview and offered appropriate comfort while allowing the person time. The recorded interview only recommenced if/when the participant indicated that they were ready to continue. 

In the event of a participant becoming distressed, interviewers were prepared to provide contact details for an independent health professional (social worker or psychologist). 

Also, provide a profile of the sites that does not disclose who they are. The profile of the sites (as summarised in Table 1) has been approved by both sites and their ethics committees. Neither site nor the ethics committees expressed any concerns about the amount of detail provided about the participating services. 

We feel that the information provided in Table 1 provides important context around the location and operating conditions of both sites, especially for international readers. However, I have added a footnote to Table 1 to clarify a point that may have been misleading. 

Finally, we note that it is difficult to maintain true anonymity of the services when two of the co-authors work for the services, the name and location of which must be stated in their author affiliation.

8. Why were the interviews done over a long timeframe? Could the reasons be averted?

8: The interviews for the detailed service studies were conducted over two years, between September 2017 and December 2018. However, as explained in the paper, we included a small number of interviews that were conducted as part of the original service interviews between March and October 2015. We have modified the text as follows to clarify this point (new text in italics): 

The interviews for the detailed service studies were conducted between September 2017 and December 2018 by four female researchers. Two of the researchers conducting interviews were Aboriginal women, both of whom have clinical backgrounds in cancer, and one of whom is an experienced researcher. The two non-Indigenous interviewers both have clinical backgrounds and over twenty years’ experience with collaborative research into improving Indigenous health outcomes. We also included four staff interviews from the original service interviews conducted between March and October 2015; these were conducted via telephone by one non-Indigenous female researcher with five years’ experience in Indigenous health research.

Three staff were interviewed twice, initially in 2015 as part of the original service interviews, and again in 2017 or 2018 as part of the detailed service study. The original interviews with these participants were included in the analysis as they offered rich additional insights into their service’s workforce approach. An interview with one key staff member was included from the 2015 study because they were on long leave, and therefore unavailable for interview during the detailed service study.

To work respectfully in this space conducting research with health service providers and with Aboriginal staff members, the timeline necessarily was driven by the priorities, capabilities and resources of the health services and health professionals that we were working with. Health services have multiple priorities, with research not the highest of these. Reasons for delays in this study were multifactorial and included: 

• Time taken to get to know, work with and gain the trust of the health service personnel, particularly the managers of the Indigenous Health Units. 

• Gaining permissions and ethics approvals

• Waiting for internal restructure of health services

• Changeover of personnel

These reasons were mostly outside of our control and could not easily have been averted. 

9. Findings: Currently, the paper needs a deeper analysis and reporting of the data findings. Currently, as it stands, it reads like a carefully crafted government report: good things at the start and one section on things to build upon. This paper needs the data in the final section, gaps and continuous improvement, to be located under the other headings. For example, paragraph two could go under the section on employment.

The section read overly positive. The reading of the literature does not stack up to this research paper.

9: Firstly we’d like to stress that the purpose of this study was not to describe the challenges experienced by these services, but to take a strength-based approach to describing how these two services supported their Indigenous workforce and provided quality of care for Indigenous people, in order to highlight how other services could learn from these high-performing services. While it is true that we could make more of the challenges that exist in this space, they have been written about extensively elsewhere. What is needed is to identify what works to overcome the evident challenges experienced by Aboriginal people. What we have stressed is that these are not just high performing cancer services but cancer treatment centres which are supported by highly functional health centres attuned to the needs of Aboriginal people.

Furthermore, we want to stress that it was a long and considered process to identify the two services that participated in this study. One of the overarching purposes of this study was to identify the cancer services providing the best care and support for Aboriginal patients and their families. To this end, we conducted a national survey of all cancer providers, with follow up interviews with a smaller selection of services. From our national survey and interviews, only four services were deemed to be providing sufficiently high levels of culturally safe care and support to Aboriginal people to be suitable candidates for this case study. One service will be reported elsewhere. After conducting the case studies, one service was excluded from analysis and write up because it was not considered by the research team to be high performing with respect to specific attention on care of Indigenous people or of their Indigenous workforce. 

10. Discussion: the discussion just repeats the findings with limited interrogation of the data relating to the literature. It is hard to believe that these two agencies do not have challenges.

10: We have expanded on the Discussion, with some additional interrogation of the data relating to the literature. We have also removed some sentences that were just repeating the findings. However, again we wish to stress that the purpose of this study was not to describe the challenges experienced by these services, but to take a strengths-based approach. 

---

Reviewer 2: 

1. The authors have presented a well articulated and presented paper defining the key role of workforce to support improved health for people receiving cancer care, which promotes the qualitative experiences of an Indigenous Health Workforce, and non-Indigenous colleagues, and explores the operating environments of the workforce within the wider health service.

This is an important and current topic, to improve the care experiences within mainstream health services, and reports on the outcomes of significant service investments to embed cultural safety and respect. I congratulate the health services who were profiled, and the research team who express this work.

1: We thank the reviewer for these comments. 

2. Abstract Intro: sentence 1 is negative framing. Perhaps swap the first 2 sentences of the abstract. I note the Intro first sentence is strength based, and the remainder of the paper is strengths based.

2: Thank you for pointing this out. It was our intention that this article be strength-based. The first sentence of the Abstract has been re-written as follows to be more strength-based: 

Improving health outcomes for Indigenous people by strengthening the cultural safety of care is a vital challenge for the health sector, both in Australia and internationally.

3. Discussion: I note absence of discussion against other publications in Australia or overseas on Cancer care workforce - this would help to see if these local strategies were aligned to best practices elsewhere, or indeed are likely to be internationally leading innovations.

3: We have added the following discussion of the cancer care workforce into the Discussion (new content indicated below in italics): 

Although strong health service leadership, mission statements and policies are vital for the support of the Indigenous workforce, there are numerous examples in the literature of managerial directives and service-wide policies and procedures not being translated into practice “on the ground” [13, 61, 62]. However, it was these cancer services’ respect for and support of their Indigenous staff that led us to examine the affiliated health services more broadly. The genuine valuing of Indigenous staff came from the deep respect in seeing the advantages that they brought to the patient care role and was manifest in their automatic inclusion in patient care team meetings, not at observers but as consultants to senior health professional staff. This contrasts with the literature, where although there have been calls both in Australia and internationally to create a more diverse cancer workforce capable of providing culturally safe cancer care [63-65], we found few reports of cancer services successfully supporting or developing an effective Indigenous health workforce. Successful initiatives included two Indigenous patient navigator (IPN) programmes in the USA and a pilot IPN programme in Australia [66, 67] and an Indigenous palliative care workforce education program in Australia [68]. 

4. Discussion: Many recommendations for health services are consistent with Australian Commission on Safety and Quality of Health Care, including care for Aboriginal and Torres Strait Islander peoples. I would recommend these publications be cited. I note the appropriate citation of strategic papers for Australian Indigenous workforce.

4: We have now cited and quoted from both the Australian National Safety and Quality Health Service (NSQHS) Standards and the NSQHS Standards User guide for Aboriginal and Torres Strait Islander health throughout the Discussion, including in the Recommendations. 

5. Discussion: Please address the comment about Indigenous peoples within the medical workforce:

Many professional and service roles were described, with a large emphasis on ILO, yet there was no reference to the importance of Indigenous people in medical pathways (as junior doctors, general practitioners and consultant oncologists and radiation oncologists or allied health. If these were not identified in the thematic analyses, perhaps this can be explored in the discussion.

5: There was minimal mention of Indigenous people in other clinical roles in the data. We have added the following paragraph to the Discussion, to explore this: 

Although the importance of ILOs came through strongly in the interviews, there was very little mention of Indigenous staff in other clinical roles. The Indigenous staff interviewed were employed in a variety of roles (three ILOs, two in management positions, two nurses and one in administration), however references to Indigenous staff in other roles (such as Indigenous nurses, social workers and a physiotherapist) were brief and incidental. There was no mention of Indigenous doctors. This is despite the known importance of Indigenous people filling a variety of clinical roles, and the almost two-decade long push to increase the number of Indigenous doctors [46, 55, 56]. There are a number of possible reasons for this. Firstly, ILOs are in identified roles, which makes them more evident within the organisation than Indigenous colleagues in nursing, medicine and allied health, who may prioritise identifying in their health professional role over that of being Indigenous. As stated in the introduction, Indigenous Australians are under-represented in the health workforce, with large disparities between rates of Indigenous and non-Indigenous employees for every health profession, including nurses (in 2015, 1.1% of nurses and midwives identified as Indigenous) and doctors (0.5% of doctors identified as Indigenous) [57, 58]. Furthermore, the focus of this study was on cancer services, with specific questions about the ILOs who worked in oncology. If we had interviewed additional staff from other areas of the health service, we may have collected more data on Indigenous people in other clinical roles.

6. Strengths and Limitations of methods- I suspect there was saturation of themes within the qual interviews- can this be confirmed. 

6: During analysis of the staff interviews we noticed that the experiences described began to replicate, suggesting that saturation may have been reached amongst this participant group. However, due to a number of factors, we were only able to interview eight Indigenous people affected by cancer. This group has huge diversity of experience so we could not claim saturation with this participant group. 

7. Do the authors regard this methodological approach as wholly satisfactory, or would they suggest modification, if this was extended to more sites in the future who may still be on a workforce innovation trajectory?

7: The original intention was for this study to utilise a mixed methods approach, and to incorporate quantitative data such as patient pathway timeframes to support the qualitative data. However, there were a number of delays with this project, which led to reduced capacity and challenges with approval processes and timeframes, which meant the quantitative component of the research was not progressed. We were also interested in exploring in more detail the links between high performing services and primary health care services, but it proved difficult to identify primary health care services with sufficient numbers of patients and experience to progress this component.

We have provided the following information in the new Limitations section: 

The original intention was for this study to utilise a mixed methods approach, and to incorporate quantitative data such as patient pathway timeframes to support the qualitative data. However, there were a number of delays with this project as well as challenges with approval processes and timeframes, which led to reduced capacity and meant the quantitative component of the research was not progressed.

Best regards,

Emma Taylor, on behalf of the authors.

---

## [Decision Letter · Decision Letter 1]

2 Sep 2020

"We’re very much part of the team here":  A culture of respect for Indigenous health workforce transforms Indigenous health care

PONE-D-20-06487R1

Dear Dr. Taylor,

We’re pleased to inform you that your manuscript has been judged scientifically suitable for publication and will be formally accepted for publication once it meets all outstanding technical requirements.

Kind regards,

Melissa T. Baysari

Academic Editor

PLOS ONE

Additional Editor Comments (optional):

Reviewers' comments:

Reviewer's Responses to Questions

**Comments to the Author**

1. If the authors have adequately addressed your comments raised in a previous round of review and you feel that this manuscript is now acceptable for publication, you may indicate that here to bypass the “Comments to the Author” section, enter your conflict of interest statement in the “Confidential to Editor” section, and submit your "Accept" recommendation.

Reviewer #2: All comments have been addressed

2. Is the manuscript technically sound, and do the data support the conclusions?

Reviewer #2: Yes

3. Has the statistical analysis been performed appropriately and rigorously? 

Reviewer #2: Yes

4. Have the authors made all data underlying the findings in their manuscript fully available?

Reviewer #2: Yes

5. Is the manuscript presented in an intelligible fashion and written in standard English?

Reviewer #2: Yes

6. Review Comments to the Author

Reviewer #2: I am satisfied my comments are adequately addressed. The revisions for Reviewer 1 also enhanced the paper. It was particularly important to frame the work as a case study (2 sites), and this gave reassurance to limitation of saturation data themes for few patient interviews. I recognise the challenge of interview for unwell and symptomatic clients to a detailed interview process. Thankyou for the opportunity to comment.

7. PLOS authors have the option to publish the peer review history of their article (what does this mean?). If published, this will include your full peer review and any attached files.

Reviewer #2: No

---

## [Editor Report · Acceptance letter]

11 Sep 2020

PONE-D-20-06487R1 

“We’re very much part of the team here”:  A culture of respect for Indigenous health workforce transforms Indigenous health care 

Dear Dr. Taylor:

I'm pleased to inform you that your manuscript has been deemed suitable for publication in PLOS ONE. Congratulations! Your manuscript is now with our production department. 

Kind regards, 

on behalf of

A/Professor Melissa T. Baysari 

Academic Editor

PLOS ONE